# Citicoline for the Management of Patients with Traumatic Brain Injury in the Acute Phase: A Systematic Review and Meta-Analysis

**DOI:** 10.3390/life13020369

**Published:** 2023-01-29

**Authors:** Julio José Secades, Helmut Trimmel, Byron Salazar, José Antonio González

**Affiliations:** 1Medical Department, Ferrer, 08029 Barcelona, Spain; 2Department of Anaesthesiology, Emergency Medicine and Critical Care, General Hospital of Wiener Neustadt, 2700 Wiener Neustadt, Austria; 3Faculty of Medicine and Dentistry, Danube Private University (DPU), 3500 Krems, Austria; 4Department of Neurosurgery, Hospital Militar de Quito, Quito 170102, Ecuador; 5Statistics and Operations Research Department, UPC, Barcelona-Tech, 08034 Barcelona, Spain

**Keywords:** CDP-choline, citicoline, management, traumatic brain injury, neuroprotection, meta-analysis, systematic review

## Abstract

Background: Citicoline or CDP-choline is a neuroprotective/neurorestorative drug used in several countries for the treatment of traumatic brain injury (TBI). Since the publication of the controversial COBRIT, the use of citicoline has been questioned in this indication, so it was considered necessary to undertake a systematic review and meta-analysis to evaluate whether citicoline is effective in the treatment of patients with TBI. Methods: A systematic search was performed on OVID-Medline, EMBASE, Google Scholar, the Cochrane Central Register of Controlled Trials, ClinicalTrials.gov, and Ferrer databases, from inception to January 2021, to identify all published, unconfounded, comparative clinical trials of citicoline in the acute phase of head-injured patients— that is, treatment started during the first 24 h. We selected studies on complicated mild, moderate, and severe head-injured patients according to the score of the Glasgow Coma Scale (GCS). The primary efficacy measure was independence at the end of the scheduled clinical trial follow-up. Results: In total, 11 clinical studies enrolling 2771 patients were identified by the end. Under the random-effects model, treatment with citicoline was associated with a significantly higher rate of independence (RR, 1.18; 95% CI = 1.05–1.33; I2, 42.6%). The dose of citicoline or the administration route had no effect on outcomes. Additionally, no significant effects on mortality were found, and no safety concerns were noticed. Conclusions: This meta-analysis indicates some beneficial effects of citicoline’s increasing the number of independent patients with TBI. The most important limitation of our meta-analysis was the presumed heterogeneity of the studies included. Registration: PROSPERO CRD42021238998

## 1. Introduction

Traumatic brain injury (TBI) is a acquired insult to the brain from an external mechanical force that can result in a temporary or permanent impairment or death, especially in moderate and severe cases [1], which is considered a major worldwide neurological disorder of epidemic proportions [2]. To date, there are still no FDA-approved therapies to treat any forms of TBI [2]. In 2016, there were 27.08 million (24.30–30.30 million) new cases of TBI in the world, with an age-standardized incidence rate of 369 (331–412) per 100,000 persons [3]. In the same year, the number of prevalent cases of TBI was 55.50 million (53.40–57.62 million). From 1990 to 2016, the age-standardized prevalence of TBI increased by 8.4% (7.7 to 9.2) worldwide [3]. TBI affects both elderly persons, mainly due to falls, and young individuals, because of road injuries, assault, or sports- and work-related accidents [4,5]. A high proportion of survivors from moderate or severe head injuries exhibit some grade of disability, with reduced independence [4,6]. For this reason, TBI is associated with a vast health and economic burden [7]. Permanent cost estimates ranged from $279 million to $1.22 billion, depending on the diagnostic criteria used to define TBI [7]. Therefore, it is imperative to treat TBI, with the aims of reducing severity and improving recovery [5].

It is considered that a better comprehension of the complex pathophysiology of TBI will help to decrease the frequency of TBI-associated complications. Alterations in cell membrane integrity, as well as impairments of the lipid metabolism, can lead to cell death after TBI, as in ischemic brain injuries [8,9,10]. In this sense, the importance of the role of lipids in cell signaling and tissue physiology has been demonstrated in many central nervous system (CNS) disorders and injuries related to dysregulated metabolism, as a key step in the pathophysiology of the ischemic/traumatic brain injury [8]. Choline-based phospholipids are involved in the maintenance of the structural integrity of the neuronal and glial cell membranes and are simultaneously the essential component of various biochemical pathways, such as cholinergic neurotransmission in the brain [9]. Consequently, a therapeutic approach based on the protection and the regeneration of cell membranes and on the normalization of lipid metabolism could prove beneficial [5,8] as a neuroprotective strategy. As a treatment approach, neuroprotection could be considered beneficial in patients with TBI [5,11,12] among other areas of improvement, such as intracranial pressure management, neuromonitoring, and the management of paroxysmal sympathetic hyperactivity [12].

Among the neuroprotective drugs used for the management of brain ischemia is citicoline [8]. Citicoline (cytidine 5′-diphosphocholine, or CDP-choline) is an endogenous compound involved in the biosynthesis of phosphatidylcholine, the major neuronal membrane lipid. Exogenous administration of citicoline has been shown to generate phospholipids, thus making it a neuroprotective/neurorestorative agent with an appropriate benefit–risk profile in the management of brain-ischemia-related disorders [5,13,14]. Citicoline has a pleiotropic effect on the molecular events involved in the pathophysiology of ischemic/traumatic brain injury, and it is widely used as a neuroprotective treatment of stroke and head injuries and the sequelae of both diseases [14]. In 2012, the Citicoline Brain Injury Treatment Trial (COBRIT) was published. This trial did not reveal an improvement in functional or cognitive status compared with placebo [15]. In any case, COBRIT was subject to a number of limitations, such as the extremely low adherence and the atypical oro-enteral administration of citicoline. The latter is not approved for TBI in any country, has not yet been scientifically studied, and, among other things, is not suitable for many of the patients included in the study [5,16]. Additionally, a meta-analysis performed in 2014 by Secades [17] showed that citicoline could have a positive impact on the rates of independence among patients following TBI, whereas the meta-analysis of El Sayed et al. [18] reported a neutral effect of citicoline on the Glasgow Outcome Scale/Glasgow Outcome Scale extended (GOS/GOSe) score, cognitive performance, and survival. Recently, an exhaustive narrative review on the role of citicoline in TBI has been published [5].

Therefore, given the available evidence, the role of citicoline in patients with TBI should be further investigated and clarified, especially after the controversy arising from the neutral results of the COBRIT. The aim of this study was to assess the benefits and hazards of therapy with citicoline in patients with TBI through a systematic review and meta-analysis of comparative trials.

## 2. Materials and Methods

This systematic review was conducted according to the methodological standards of the Cochrane Collaboration [19] and was based on a protocol (CRD42021238998) registered at https://www.crd.york.ac.uk/PROSPERO (accessed on 17 August 2022) [20]. The report followed the Preferred Reporting Items for Systematic Review and Meta-analysis (PRISMA) statement guidelines [21].

### 2.1. Study Selection Criteria

Eligible studies included randomized controlled trials (RCTs), comparative studies, and cohort studies of male and female patients of any age with a diagnosis of TBI and treatment starting in the first 24 h after injury and included the reporting results for independence or functional outcome. The trials included analyses of patients with complicated mild (defined as patients with a baseline Glasgow Coma Scale (GCS) score of 13–15 with some lesions on the CT scan), moderate (GCS 9–12), and severe TBI (GCS 3–8), but not mild TBI. The main reason for including complicated mild patients was because independence results for this category of patients were included in the COBRIT publication [15]. This systematic review comprised all studies in which the active agent was citicoline, irrespective of whether it was compared with another active treatment.

### 2.2. Search Methods

We searched OVID-Medline, EMBASE (access through OVID), Google Scholar, the Cochrane Central Register of Controlled Trials (CENTRAL) (the Cochrane Library, latest issue), and the US National Institutes of Health’s ClinicalTrials.gov website, from inception to week three of January 2021, using appropriate controlled vocabulary and free search terms. Search strategies for OVID, MEDLINE, and EMBASE are detailed in Appendix A. For the other searches, the keywords CDP-choline, citicoline, traumatic brain injury, head injury, and craniocerebral trauma were used. Additionally, the reference lists from eligible studies, other reviews on this topic, and the Ferrer bibliographic database were screened to identify relevant studies. No restrictions on language, publication date, or publication status were applied. We did not search the gray literature.

### 2.3. Study Selection and Data Extraction

One author (J.J.S.) reviewed the abstracts of the articles retrieved in the search. Full-length papers for any abstract that we thought met the inclusion criteria were obtained. Three authors (J.J.S., H.T., B.S.) reviewed all the retrieved papers to identify those trials meeting the inclusion criteria for the review. One review author (J.J.S.) initially extracted the efficacy data from eligible trials that were then independently confirmed by H.T. and B.S. Discrepancies were resolved by discussion with the other authors (H.T., B.S., J.A.G.) and by referencing the original report. J.A.G. performed all the statistical analyses.

### 2.4. Risk of Bias Assessment

We used the Cochrane Collaboration’s tool to assess the risk of bias in the included studies [19]. We scored the risk of bias by using the RoB2 tool v.7 [22] for randomized trials and the ROBINS-I [23] for nonrandomized studies.

### 2.5. Data Analysis

The primary efficacy measure was independence at the end of the scheduled clinical trial follow-up. If available, we used the GOS/GOSe for this measure. In studies lacking the GOS/GOSe measurement, we used the most comprehensive measure of disability or handicap available from the study for the classification of patients as dependent or independent, defining *independence* as the ability to perform almost all the activities of daily life without the assistance of another.

We applied dichotomous outcomes in the statistical meta-analytic methods, and a weighted estimation (with inverse-variance weights) was used. For the GOS, scores of 4 or 5 were considered as good outcomes (independence), and for GOSe, 7 or 8. The risk ratio (RR) was used to test for the proportional treatment effects of citicoline. The odds ratio (OR) and risk difference (RD) were used in complementary analysis.

The formal meta-analysis was performed by using metafor (version 2.4-0), a meta-analysis package for R [24]. Given that we assumed heterogeneity with respect to the studies performed over 4 decades, the main analysis used the random-effects model to obtain a 95% confidence interval (CI) estimate for the effects of citicoline compared with controls. We chose the REML (restricted maximum likelihood estimator) method, that is, the default in the package; confidence intervals are based on a standard normal distribution. The strength of the body of evidence was assessed by using the grading of recommendations assessment, development, and evaluation (GRADE) score [25]. The results of the meta-analysis were presented as forest plots, with studies listed in order of age, oldest first. In addition to unadjusted random- and fixed-effects models, as subgroups data were analyzed according to the dose (trials with doses higher than 2 g/day vs. 2 g/day or lower) and the administration route (parenteral vs. only oral/enteral) because these study-level variables included in the model might account for part of the heterogeneity in the effects. We also performed a sensitivity analysis restricted to RCT studies.

## 3. Results

### 3.1. Study Selection

The search results and the decisions made during the eligibility process are displayed in the PRISMA flowchart (Figure 1). The search provided 2460 records. Another 96 records were identified through the search of the Ferrer bibliographic database. The removal of duplicates left a total of 2500 references. After a review of the citations, the abstracts, and the full papers (when available), 96 records were initially screened. In total, 59 references were excluded as they corresponded to animal studies or reviews. Consequently, in the end, 37 full-text clinical studies were assessed for eligibility (Figure 1). Among these clinical studies, only 11 fulfilled the criteria to be included in the qualitative synthesis [15,26,27,28,29,30,31,32,33,34,35]. Among the 26 studies not selected for the meta-analysis, six were noncomparative studies [36,37,38,39,40,41], six were not in the acute phase of TBI [42,43,44,45,46,47], six had unavailable independence data [48,49,50,51,52,53], three compared different combinations of drugs [54,55,56], two analyzed different types of patients (including few with TBI) [57,58], one assessed cervical trauma [59], one assessed mild TBI [60], and one study reported no clear results, with only 10 patients per arm [61].

### 3.2. Study Characteristics

The 11 included studies comprised two cohort studies [31,35], seven RCTs comparing citicoline with placebo or a control group [15,26,27,28,32,33,34], one RCT comparing citicoline with meclofenoxate [29], and another RCT comparing citicoline with piracetam [30]. All the studies assessed the effect of citicoline on the recovery of patients with complicated mild, moderate, or severe head injury. The oldest study was published in 1978, and the most recent study was published in 2018; thus, there was a gap of 40 years between the first and last studies. As stated above, this may be a source of heterogeneity owing to improvements in the management of TBI during this period, and it justifies an analysis based on the random-effects model. The studies included a total of 2771 patients, in whom citicoline was administered at doses ranging from 300 mg to 6 g. The duration of treatment varied from 10 to 90 days. The drug was administered intravenously in six studies [26,29,30,31,33,35], intravenously or intramuscularly in two studies [27,28], intravenously followed by oral administration in one study [32], and orally in two studies [15,34]. All the studies included the rate of independence, albeit at different times of evaluation (Table 1). The PEDro score [62] of the trials ranged from 7 to 11, with an average of 9.6 (Table 2).

Figure 2 shows the methodological quality of the included randomized studies, based on the RoB2 tool, with a low risk of bias. According to the ROBINS-I assessment tool for nonrandomized studies, the two studies [31,35] included in this meta-analysis can be considered to have a low or moderate risk of bias for all domains. One major difference between the trials was the standard of care applied, which changed over time.

### 3.3. Synthesis of the Results

The effect estimates and the confidence intervals (CIs) are presented as a forest plot. All the results were directly reported, as obtained from the original publication. The administration of citicoline was associated with a significantly higher rate of independence (RR = 1.18; 95% CI = 1.05–1.33; I2 = 42.6%) (Figure 3). Complementarily, we performed OR and RD analyses, and in both cases, the results were congruent with the RR obtained (OR = 1.56; 95% CI = 1.15–2.12 (Figure 4); RD = 0.12; 95% CI = 0.04–0.19 (Figure 5)). Thus, the probability of presenting the favorable event was 18% higher with the intervention; alternatively, citicoline increased by 0.12 points with respect to that of a favorable event without the intervention. The I^2^ heterogeneity indicator obtained (42.6%) was not large but was considerable, as described in the funnel plot (Figure 6); nonetheless, we also estimated the effect of citicoline under the fixed-effects model, with similar results (Table 3). 

None of the adjusted analyses yielded a significant difference, and thus, we were unable to find a relationship between the outcome and either the dose of citicoline or the route of administration. Table 3summarizes the results obtained. Note that the significant effect was maintained when we analyzed only the RCTs (RR = 1.16; 95% CI = 1.01–1.33; I2 = 39.4%). According to the results, we can categorize the evidence as moderate certainty (GRADE). There was no difference in mortality rates in the randomized studies. Only in the study of Trimmel et al. [35] was a significant reduction in mortality for patients with severe TBI treated with citicoline described. Additionally, none of the included studies reported any serious safety problem associated with citicoline.

## 4. Discussion

This meta-analysis showed that in nearly 2800 patients with acute-phase TBI, treatment with citicoline was associated with a significant improvement in the level of independence, with a moderate level of evidence according to GRADE.

Given the considerable disease burden associated with moderate to severe TBI, there is a need to improve recovery in affected patients. Although the treatment for TBI has improved in recent years, mortality and disability rates remain high [3,4,9,10]. Inflammation, alterations in cell membrane integrity, and the impairment of phospholipid metabolism have all been implicated in the pathophysiology of TBI [5,8,9,10,14]. Importantly, many studies have shown that citicoline has neuroprotective and neurorestorative properties, including the following: the normalization or stabilization of damaged neuronal cell membranes (i.e., phospholipid content and function, ion exchange); the restoration of some enzymatic activities; a reduction in the generation of damaging free fatty acids and free radicals; the improvement of neurotransmission and cerebral metabolism; anti-inflammatory and antioxidant properties; the enhanced integrity of the blood–brain barrier; the accelerated absorption of brain edema and the decreased volume of ischemic lesions; the inhibition of apoptosis; and the enhancement of neurorepair and neuroplasticity properties [5,8,9,10,14,63,64,65,66]. Thus, given its biochemical, pharmacological, and pharmacokinetic characteristics, citicoline should considered as a potentially useful drug for the treatment of patients with TBI [5,14,17].

Nevertheless, the COBRIT did not show any improvement in outcomes in this population [15]. The COBRIT has been the largest study performed with citicoline in TBI patients, but there are relevant methodological issues that question the validity and applicability of the results obtained in the study. The first point to consider is the financing of this study; the study was an independent study, financed by the US National Institute of Health, with a limited budget. A relevant point to consider is the sample size calculation. The authors chose an OR of 1.4 as the effect of the treatment, when in the most recent publications, the size of the effect of citicoline was 1.26 in acute ischemic stroke patients, a less heterogeneous pathology than TBI. The sample size was likely calculated on the basis of the number of patients that could be afforded, and then the OR of the treatment was established accordingly, instead of basing it on the real effects of the drug. With a more conservative and realistic OR of 1.2 or less, the sample size should be much higher and would likely have been unaffordable for the authors. Another questionable point to consider is that the authors mixed different populations, confusing mild, moderate, and severe TBI. The pathophysiology, localization, and trajectory for recovery can be quite different among these different groups. To avoid this issue, the authors should have used a randomized, matched sample design. This mixing of lesion severity is a clear source of heterogeneity and would have to be considered an important confounding factor in the analysis and interpretation of the data. Another point is the atypical oro-enteral administration of citicoline used in this trial, which is not approved in any country, has not previously been scientifically tested, and is not appropriate for many of the patients enrolled in the study, particularly in moderate and severe cases. However, the most controversial point is the extremely poor compliance with the treatment. Only 44.4% compliance for patients having taken more than 75% of the medication expected is clearly insufficient and needs further elaboration in the interpretation of the results. Not receiving the active treatment is not the same as not receiving the placebo, in terms of the standard of care being received. This means that fewer than half of the patients received something close to a therapeutic dose of citicoline. Thus, the COBRIT is not the definitive study on citicoline, especially when the methodological confounds just described are taken into consideration [5,14,16]). As a result, the evidence provided by the study must be considered controversial [5,16].

In this context, meta-analyses could prove to be very helpful in clarifying the role of citicoline in clinical practice. In 2014, a previous meta-analysis based on 12 clinical studies enrolling 2706 patients with mild to severe TBI treated in the acute phase with/without citicoline showed a significant increase in the rates of independence with citicoline (OR, 1.815 (95% CI, 1.302–2.530) under the random-effects model vs. 1.451 (95% CI, 1.224–1.721) under the fixed-effects model) [17]. These results are in line with those in this study. In contrast, another meta-analysis found neutral effects for citicoline in the treatment of patients with TBI, although it is noteworthy that this meta-analysis was restricted to studies published in English, with only 1355 patients for the GOS outcome, 1,291 patients for the assessment of cognitive performance, and 1037 patients for the assessment of survival [18]. Therefore, the sample size in this meta-analysis was too limited to find a beneficial impact of citicoline in the study population, owing to the English language restriction, which is a well-known source of heterogeneity [5].

While our study analyzed only the neuroprotective efficacy of citicoline among patients following TBI, clinical trials and real-life studies have confirmed its excellent safety profile [5,14,17]. The neuroprotective effects of citicoline should not be limited to patients with TBI but instead could also be extended to patients with other neurological diseases [14], as some studies have suggested, including COVID-19-related cognitive complications, multiple sclerosis, and dementia [67,68,69].

The main limitation of our meta-analysis was the presumed heterogeneity of the included studies. This was the result of marked improvements in the management of patients with TBI over 4 decades. As shown in Figure 6, a funnel plot suggests that the old, small studies may be overestimating the effect size, although the number of these studies is too few to be conclusive. In addition, our meta-analysis was the largest performed to date and analyzed the level of independence, an outcome that was well defined in the included studies. In any case, our results are quite relevant because no new large-scale clinical trials with citicoline for this indication are expected.

The initial extraction of the data was performed by only one author (J.J.S.) because he was the author of a recent narrative review on the effects of citicoline on TBI [5]. Despite that, the initially extracted data were confirmed by the other clinical authors (H.T. and B.S.).

## 5. Conclusions

In summary, with regard to the level of independence, our meta-analysis provided some evidence of the benefits of citicoline in combination with the standard of care in the management of patients with complicated mild, moderate, and severe TBI. This benefit could be independent of the dose used and of the administration route (oral or parenteral).

## Figures and Tables

**Figure 1 life-13-00369-f001:**
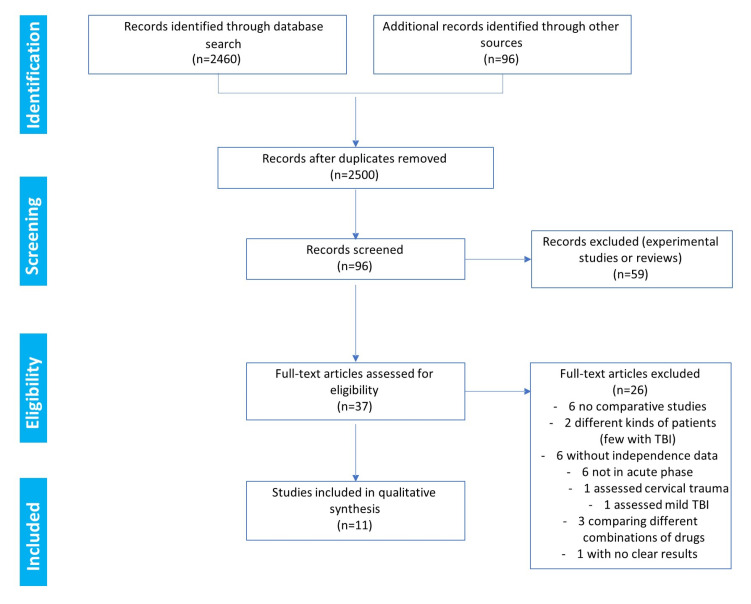
Eligibility: PRISMA flowchart.

**Figure 2 life-13-00369-f002:**
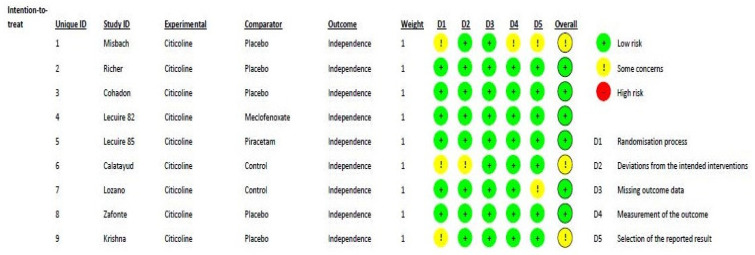
Risk of bias distribution diagram.

**Figure 3 life-13-00369-f003:**
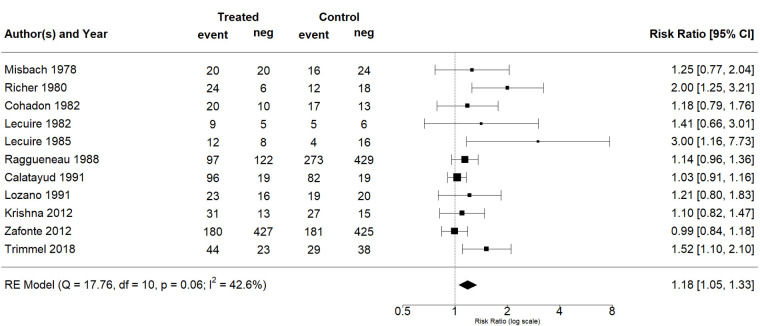
Forest plot of the meta-analysis, based on the random-effects model (risk ratio).

**Figure 4 life-13-00369-f004:**
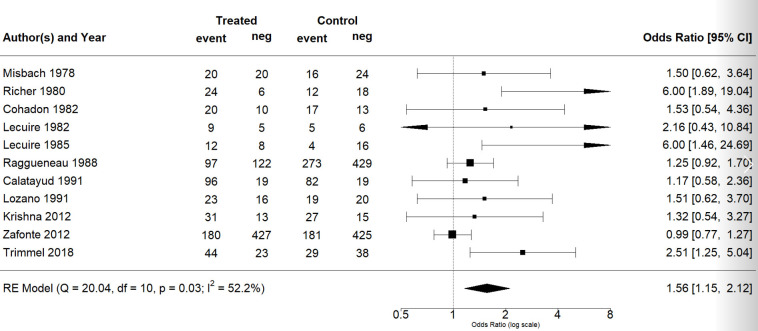
Forest plot of the meta-analysis: complementary analysis (odds ratio).

**Figure 5 life-13-00369-f005:**
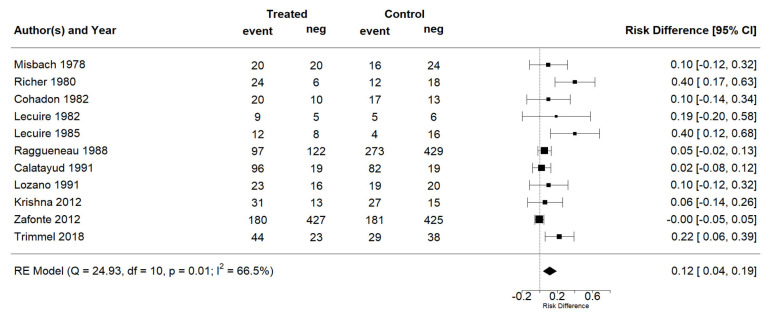
Forest plot of the meta-analysis: complementary analysis (risk difference).

**Figure 6 life-13-00369-f006:**
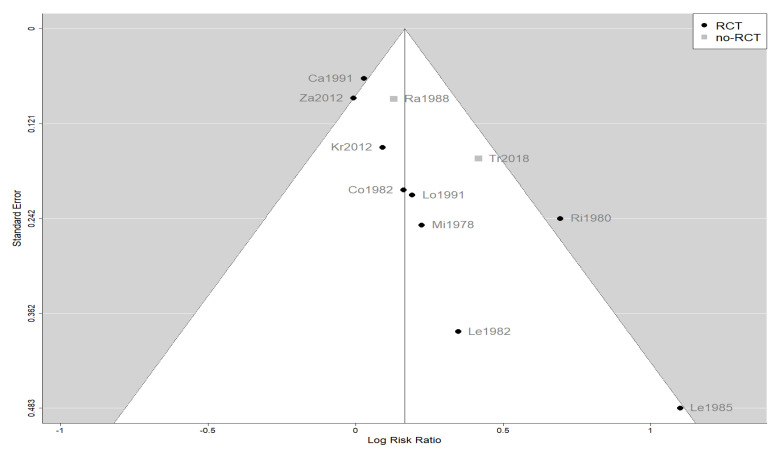
Funnel plot of the meta-analysis showing the heterogeneity among the studies included in the meta-analysis.

**Table 1 life-13-00369-t001:** Summary of the studies evaluating the efficacy of citicoline in TBI included in the present analysis.

Study	Year	n	Severity	Control	Doses	Outcome
Misbach et al. [26]	1978	80	Moderate to severe	Placebo	300 mg IV × 14 d	Recovery rate at 14 d
Richer et al. [27]	1980	60	Severe	Placebo	750 mg IV or IM × 20 d	GOS (like) at 3 m
Cohadon et al. [28]	1982	60	Severe	Placebo	750 mg IV or IM × 20 d	GOS (like) at 2 m
Lecuire et al. [29]	1982	25	Moderate to severe	Meclofenoxate	750 mg IV × 10 d	Global recovery at 10 d
Lecuire [30]	1985	40	Moderate to severe	Piracetam	750 mg IV × 10 d	Global recovery at 1 m
Raggueneau et al. [31]	1988	921	Severe	Control	500–750 mg IV × 20 d	GOS at 1 m
Calatayud et al. [32]	1991	216	Moderate to severe	Control	3–4 g IV × 4 d 2 g po × 26 d	GOS at 3 m
Lozano [33]	1991	78	Moderate to severe	Control	3–6 g IV × 1	GOS at 1 m
Zafonte et al. [15]	2012	1070	Mild complicated, moderate, and severe	Placebo	2 g po × 90 d	GOSe at 6 m
Krishna et al. [34]	2012	87	Moderate to severe	Placebo	2 g po × 60 d	GOS at 90 d
Trimmel et al. [35]	2018	134	Severe	Control	3 g IV × 21 d	GOSe at 6 m

**Table 2 life-13-00369-t002:** PEDro scores of the studies.

Study	C1	C2	C3	C4	C5	C6	C7	C8	C9	C10	C11	TOTAL
Misbach et al. [26]	Y	Y	Y	Y	Y	Y	Y	Y	Y	Y	N	10
Richer et al. [27]	Y	Y	Y	Y	Y	Y	Y	Y	Y	Y	Y	11
Cohadon et al. [28]	Y	Y	Y	Y	Y	Y	Y	Y	Y	Y	Y	11
Lecuire et al. [29]	Y	Y	Y	Y	Y	Y	Y	Y	Y	Y	Y	11
Lecuire [30]	Y	Y	Y	Y	Y	Y	Y	Y	Y	Y	Y	11
Raggueneau et al. [31]	Y	N	N	Y	Y	N	N	Y	Y	Y	Y	7
Calatayud et al. [32]	Y	Y	Y	Y	Y	N	N	Y	Y	Y	Y	9
Lozano [33]	Y	Y	Y	Y	Y	N	N	Y	Y	Y	Y	9
Zafonte et al. [15]	Y	Y	Y	Y	Y	Y	Y	Y	Y	Y	Y	11
Krishna et al. [34]	Y	Y	Y	Y	Y	Y	N	Y	Y	Y	N	9
Trimmel et al. [35]	Y	N	N	Y	Y	N	N	Y	Y	Y	Y	7

C1: eligibility criteria were specified; C2: subjects were randomly allocated to groups; C3: allocation was concealed; C4: the groups were similar at baseline regarding the most important prognostic indicator; C5: there was blinding of all subjects; C6: there was blinding of all therapists who administered the therapy; C7: there was blinding of all assessors who measured at least one key outcome; C8: measures of at least one key outcome were obtained from more than 85% of the subjects initially allocated to groups; C9: all subjects for whom outcome measures were available received the treatment; C10: the results of between-group statistical comparisons are reported for at least one key outcome; C11: the study provides both point measures and measures of variability for at least one key outcome. Y = Yes; N = No.

**Table 3 life-13-00369-t003:** Summary of results *.

Model	No. of Studies	RR	95% IC	I^2^	τ^2^
Main analysis	11	1.18	1.05, 1.33	42.6%	0.014
Fixed effects	11	1.11	1.03, 1.20	43.7%	-
Only RCT	9	1.16	1.01, 1.33	39.4%	0.015
With route as moderator	Parenteral 9 Oro-enteral 2	1.25 1.03	1.08, 1.45 0.81, 1.31	42.8%	0.017
With dose as moderator	<2gr. 8 >2 gr. 3	1.21 1.18	1.02, 1.43 0.94, 1.49	47.4%	0.022

* All analyses were made as random effects, unless otherwise indicated. **τ^2^**: estimation of between-study variance.

## Data Availability

All the papers included in this meta-analysis are available upon request.

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
