# Peer review of "Citicoline for the Management of Patients with Traumatic Brain Injury in the Acute Phase: A Systematic Review and Meta-Analysis"

_life, 2023, doi:10.3390/life13020369_

Round 1
Reviewer 1 Report
The authors assessed the benefits and hazards of therapy with citicoline in patients with TBI through a systematic review and meta-analysis of comparative trials.
The paper is generally well written.
The design and interpretation of the results are proper within the scope of the data.
Author Response
Thanks for your comments
Reviewer 2 Report
Dear Author,
This is an interesting effort.
The major limitation is with the studies included as these are too old and it can be easily extrapolated that management of the TBI patients is vastly changed.
Also, the included studies have mix of severe, moderate and mild injuries it will make the sample quite heterogenous and thus the Comparision and prediction of the outcomes difficult
Also, there is no new conclusion form the presented findings
Author Response
Thanks for your comments. We agree the major limitation is the gap between studies and for that reason we decided to perform the random effects analysis. We choose to include all the studies independently of the time to avoid a possible selection bias. We would like to focus only on moderate and severe TBI but we only have the paper of the COBRIT trial that includes mild-complicated injuries and the GOSe data were not subclassified in base on severity
The present findings only wants to show that there is some effect of the drug in the management of TBI as the drug is commercialized in several countries.
Reviewer 3 Report
The authors in this systematic review and meta-analysis of comparative trials aimed to describe the benefits of citicoline in patients with TBI. This paper is well written and presentation is very good.
Author Response
Thanks for your comments
Round 2
Reviewer 2 Report
Still there are not many articles to support the evidence.
It will be better to modify as scoping or narrative review.
Author Response
Thanks for your comments
This study could be considered as complementary to a previous narrative review published recently (Pharmaceuticals (Basel). 2021 Apr 26;14(5):410. doi: 10.3390/ph14050410)
We agree to the limitations of the study, but we inluded all the information provided and with 2771 patients analyzed the information could be of some interest, as the drug is still used in the management of TBI in countries such Austria
Thus, we consider the results of some interest for the clinical practice